# Clomifene and Assisted Reproductive Technology in Humans Are Associated with Sex-Specific Offspring Epigenetic Alterations in Imprinted Control Regions

**DOI:** 10.3390/ijms231810450

**Published:** 2022-09-09

**Authors:** Dillon T. Lloyd, Harlyn G. Skinner, Rachel Maguire, Susan K. Murphy, Alison A. Motsinger-Reif, Cathrine Hoyo, John S. House

**Affiliations:** 1Center for Human Health and the Environment, North Carolina State University, Raleigh, NC 27606, USA; 2Division of Intramural Research, National Institute of Environmental Health Sciences, National Institutes of Health, Department of Health and Human Services, Research Triangle Park, NC 27709, USA; 3Bioinformatics Research Center, North Carolina State University, Raleigh, NC 27607, USA; 4Department of Biological Sciences, North Carolina State University, Raleigh, NC 27607, USA; 5Department of Obstetrics and Gynecology, Duke University Medical Center, Duke University, Durham, NC 27701, USA

**Keywords:** differentially methylated regions, adverse offspring outcomes, infertility

## Abstract

Children conceived with assisted reproductive technology (ART) have an increased risk of adverse outcomes, including congenital malformations and imprinted gene disorders. In a retrospective North Carolina-based-birth-cohort, we examined the effect of ovulation drugs and ART on CpG methylation in differentially methylated CpGs in known imprint control regions (ICRs). Nine ICRs containing 48 CpGs were assessed for methylation status by pyrosequencing in mixed leukocytes from cord blood. After restricting to non-smoking, college-educated participants who agreed to follow-up, ART-exposed (*n* = 27), clomifene-only-exposed (*n* = 22), and non-exposed (*n* = 516) groups were defined. Associations of clomifene and ART with ICR CpG methylation were assessed with linear regression and stratifying by offspring sex. In males, ART was associated with hypomethylation of the *PEG3* ICR [β(95% CI) = −1.46 (−2.81, −0.12)] and hypermethylation of the *MEG3* ICR [3.71 (0.01, 7.40)]; clomifene-only was associated with hypomethylation of the *NNAT* ICR [−5.25 (−10.12, −0.38)]. In female offspring, ART was associated with hypomethylation of the *IGF2* ICR [−3.67 (−6.79, −0.55)]. Aberrant methylation of these ICRs has been associated with cardiovascular disease and metabolic and behavioral outcomes in children. The results suggest that the increased risk of adverse outcomes in offspring conceived through ART may be due in part to altered methylation of ICRs. Larger studies utilizing epigenome-wide interrogation are warranted.

## 1. Introduction

In the United States, increased child-bearing age and infertility rates are commensurate with increased prevalence of the birth of offspring conceived with the use of ovulation drugs or assisted reproductive technology (ART) [1,2]. Compared with spontaneously conceived singleton births, singleton births conceived with ART are associated with increased risk of multiple adverse maternal and offspring outcomes, including preterm delivery, perinatal mortality, small for gestational age, antepartum hemorrhage, congenital anomalies, maternal pregnancy hypertensive disorders, and gestational diabetes [3,4]. Pregnancies resulting from ART have also been associated with a 30–40% increase in birth defects compared with spontaneous conception [3,4,5,6,7,8,9]. Although epigenetic modifications such as the control of gene expression through DNA methylation are hypothesized to mediate some of these associations, mechanistic insights linking ART to adverse offspring and maternal outcomes remain elusive.

Data from murine models and humans suggest the timing of many ART procedures may perturb the establishment of proper methylation marks during oocyte maturation or the re-establishment of other methylation marks following the demethylation wave occurring soon after fertilization. Specifically, ART may interfere with genomic imprinting—an epigenetic process established prior to germ layer specification and stably maintained in somatic tissues throughout life [10,11]. In this process, one parental allele is silenced by the methylation of CpGs in parent-of-origin-specific differentially methylated regions (DMRs) that regulate monoallelic expression of multiple genes, known as imprint control regions (ICRs) [12,13,14]. Aberrant methylation of ICRs during gametogenesis or embryogenesis may lead to life-long alterations in gene expression and multiple human congenital disorders. For example, Silver–Russell syndrome and Beckwith–Wiedemann syndrome are the result of errors during ICR establishment [15]. 

ART has been associated with aberrant methylation of known ICRs (in placenta) and poor placental or fetal growth outcomes in both humans and mouse models [10]. In mice, superovulation induced by drugs such as clomifene has been associated with the disruption of maternal imprint marks in *SNRPN*, *PEG3*, and *KCNQ10T1* and paternal imprint marks in *H19* [16]. In mice, ART treatment resulted in small changes in methylation of control-region CpGs for *H19*, *IGF2R*, and *PEG3*, altered expression of these genes, and intrauterine growth restriction (IUGR) [10,17,18,19]. In humans, multiple studies have suggested placental epigenetic perturbation in offspring conceived through ART in imprinted genes that include *H19*, *MEG3*, *PEG3*, *PLAGL1*, and *PEG10* [10,20,21,22,23].

*H19* is maternally expressed and codes for a noncoding RNA that functions as a tumor suppressor [24] and is associated with Beckwith–Wiedemann Syndrome [25]. In contrast, the nearby paternally-expressed *IGF2* gene that encodes insulin-like growth factor 2 is involved in fetoplacental development and growth and has been implicated in both cancer and cardiovascular disease [26]. The maternally-expressed *MEG3* gene, along with *RTL1*, *MEG8/8*, and *DIO3*, are regulated by the *DLK/MEG3* ICR [27] which is also a well-studied cancer-associated locus. *PEG3* is also part of a locus housing multiple genes regulated by an ICR [28] and, similar to other imprinted regions, is associated with fetal development and fetoplacental growth [29]. *PLAGL1* is a suppressor of cell growth and is often deleted in cancer. Over-expression of *PLAGL1* in-utero is associated with transient neonatal diabetes mellitus [30]. Lastly, *PEG10* is heavily expressed in the placenta and implicated in cell-cycle progression and cancer [31]. 

Recent genome-wide methylation analyses have examined associations of ART and offspring cord-blood DNA methylation. A combined case-control study in the UK Avon Longitudinal Study of Parents and Children (ALSPAC) and Norwegian Mother, Father and Child Cohort Study (MoBa) cohorts (a total of 205 ART cases and 2439 controls) reported associations of ART with differential methylation at five CpGs, but these did not replicate in the Australian Clinical review of the Health of adults conceived following Assisted Reproductive Technologies (CHART) cohort [32]. Another assessment of genome-wide methylation in the Upstate KIDS cohort found associations of ART with decreased methylation of 11CpGs, including *IGF1R*, and hypomethylation of nine maternally imprinted genes [33]. Similar results were not found for ovulation induction alone, and only the differential methylation of the maternally imprinted *GNAS* gene persisted into childhood. The results of prior work highlight issues regarding power to determine associations of ART with methylation in offspring in genome-wide analyses and indicate a need for additional genome-wide and targeted analyses.

Unique epigenetic changes related to ART procedures, including the induction of superovulation with drugs such as clomifene, micromanipulation with intracytoplasmic sperm injection of a single sperm into the egg (ICSI), in vivo oocyte maturation, in vitro fertilization (IVF), embryo transfer (ET), and embryo culture conditions, have been reported [34,35,36]. Building on these findings, we evaluated the association of ART and clomifene use with CpG methylation at 48 CpGs in nine previously characterized ICRs.

## 2. Results

Of the 574 participants included in the analysis (Table 1; Figure 1), 27 reported the use of ART (ET/ICSI), and 22 reported the use of clomifene only in the index pregnancy. While there were no significant differences in maternal body mass index (BMI) and offspring sex between the exposed and non-exposed groups, ART-exposed participants were more likely to be older, nulliparous, and white. Offspring from exposed participants (ART and clomifene) had decreased mean gestational age and mean birth weight (*p* < 0.05; 262 days vs. 272 days and 2975 g vs. 3305 g).

### 2.1. ART ET/ICSI and Mean Offspring CpG Methylation

We evaluated ART ET/ICSI exposure, excluding participants with clomifene exposure only, and offspring DMR methylation (Figure 2). In male offspring, ART was associated with hypomethylation of the *PEG3* DMR [β(95% CI) = −1.46 (−2.81, −0.12), *p* = 0.033] and hypermethylation of the *MEG3* DMR [3.71 (0.01, 7.40), *p* = 0.049]. In female offspring, ART-IVF was associated with hypomethylation of the *IGF2* DMR [−3.67 (−6.79, −0.55), *p* = 0.022]. These associations were unaltered in a sex-adjusted model that included all participants. Several CpGs within the *PEG3* and *MEG3* DMRs were significant with a clear, consistent intra-DMR effect direction (Appendix A) whereas the mean CpG methylation for the region as a whole was not significant. In a sensitivity analysis performed by including education in the model, the magnitude of associations with the *PEG3*, *MEG3*, and *IGF2* DMRs was attenuated while the direction of association remained consistent. A sensitivity analysis performed by excluding methylation outliers did not affect these findings (Appendix A).

### 2.2. Clomifene Only and Mean Offspring CpG Methylation

We evaluated clomifene use only, excluding participants who underwent ET or ICSI ART, and offspring DMR methylation (Figure 3). In male offspring, clomifene use only was associated with mean hypomethylation of the *PLAGL1* DMR [−8.58 (−15.53, −1.63), *p* = 0.016] and the *NNAT* DMR [−5.25 (−10.12, −0.38), *p* = 0.035] (Figure 3). Appendix A displays the results of methylation changes at individual CpGs. An outlier sensitivity analysis revealed the association of clomifene exposure and male-offspring hypomethylation in the *PLAGL1* DMR was driven by a single individual, while the association for *NNAT* was unaffected (Appendix A).

## 3. Discussion

The use of ART and superovulation drugs has been associated with adverse offspring outcomes with little mechanistic understanding. A meta-analysis reporting associations of imprinting disorders with ART suggested some insights [37]. We found associations between ART and hypomethylation of the *PEG3* DMR, a gene associated with placental and fetal development and intra-uterine-growth-restriction [10]. The effect size was similar by offspring sex but was significant only for males. We also found a male-specific association of ART with hypermethylation of the *MEG3* DMR, and a female-specific association of ART with hypomethylation of the *IFG2* DMR. 

Recent work has associated ART with hypomethylation of the *PLAGL1* DMR [33]. Although our ART results also showed hypomethylation, the association was not significant. Aberrant methylation of *PLAGL1* has been implicated in placental growth and very preterm births [38]. In females, we found associations of ART and *IGF2* DMR hypomethylation. In addition to the aforementioned association of clomifene use and male-offspring *PLAGL1* hypomethylation, we also observed *NNAT* hypomethylation in male offspring conceived with clomifene. Similar patterns of methylation at these DMRs have been associated with cardiovascular disease, metabolic disease, and some cancers in adulthood [39]. These findings are consistent with the contention that DMRs are responsive to environmental influences and can serve as archives of past exposures [40,41,42,43,44] and thus the stability of their patterns of methylation makes these regions logical targets for evaluating early origins of disease [45,46,47,48]. In our study, although ART-exposed participants likely used ovulation drugs such as clomifene prior to ART, associations of ART and interrogated methylation changes were very different from women who reported the use of clomifene only, suggesting that offspring methylation differences may vary by the type of procedure/drug. 

*NNAT* is associated with the imprinting disorder Beckwith–Wiedemann syndrome, which is an imprinting disorder. A recent study of 44 ART-conceived children and controls reported placental hypomethylation of multiple imprinted genes, including *NNAT* [49]. This result was an outlier group identified by principal component analysis (*n* = 15). Our findings for *NNAT* are significant only for males whose mothers used clomifene to conceive. Importantly, the same study [49] reported highly different placental genome- wide patterns of methylation for in vitro (IVF/ICSI) procedures vs. procedures involving ovulation induction or intrauterine insemination. We also found substantial differences in methylation patterns in offspring cord blood between mothers who conceived with superovulation drugs only, and those who conceived with more invasive procedures. The imprinted gene *MEST* has been associated with male infertility as well as alterations in methylation of its ICR in both humans and mice [50,51,52,53]. We did not find significant associations of ART or clomifene exposure and offspring changes in *MEST* in offspring, but this may have been due to the small number of cases. 

Our findings are also consistent with a recent report examining the effects of ART and culture medium on methylation in buccal smears of children aged 7–8 years [54]. Based on a more global chip-based analysis of probes, investigators reported enrichment of differentially methylated probes associated with imprinted genes, including *H19*, and an increase in methylation of CpGs in the *PEG3* DMR in children exposed to ART, albeit in the opposite direction. The reasons for the differences are unclear although the probes and CpGs assessed differ.

While differential methylation at multiple sites across the genome has been reported with other environmental exposures periconceptionally in studies of adults and children [55,56], long-term adverse health outcomes in ART-conceived offspring are understudied in humans. Inconclusive data from small studies have suggested increased risk of type 2 diabetes, CVD, and metabolic syndrome (reviewed in [57]). Further, epigenetic data linking obesity to epigenetic dysregulation have been difficult to interpret for several reasons. These include the appropriateness of linking clinically accessible tissue (blood or buccal) results to tissues with etiological relevance and the fact that environmental exposures can cause temporal variability in epigenetic marks. Thus, making inferences can be difficult without temporal serial mapping of epigenetic alternations. Features of regulatory sequences of genomically imprinted genes overcome these obstacles as methylation marks at these regions are established before germ layer specification and, therefore, are similar across tissues and more temporally stable [11]. The data presented here suggest that ART and clomifene exposures that occur prior to germ layer specification are associated with altered methylation profiles at multiple loci in offspring. While a recent genome-wide scan of 450 k CpGs in the ALSPAC cohort found five CpGs with methylation values associated with ART at genome-wide significance, three of which were previously linked to human aging and disease, these findings did not replicate in the CHART cohort [32]. 

Mechanisms by which ART may alter methylation marks of putative ICRs have yet to be elucidated. It has been suggested that associated health risks may be the result of ovarian stimulation and the subsequent hormonal effects, gamete manipulation, embryo exposure to culture media, factors implicit in a couple’s need to use ART to conceive, or factors associated with non-singleton births [58,59,60]. For example, some suggest that superovulation affects maternal gene products necessary for imprint maintenance during embryonic development [61]. In mouse models, superovulation has been associated with transgenerational alterations in offspring sperm methylation [62]. Hypermethylation and gene expression of long noncoding RNA *H19* was altered in response to ART [63]. Aberrant methylation has also been suggested from the use of culture medium in IVF. The absence of protection against oxidative stress inherent with natural conception has been suggested to increase the chances of methylation errors related to imprinting [64]. Additionally, the appropriate maintenance of DNA methylation requires culture supplementation with methyl donors. Variation in folic acid supplementation may also produce suboptimal culture conditions resulting in aberrant methylation at imprinted loci with sex-specific effects [65]. This variation may be a significant contributor to methylation variation as commercial media often do not contain methyl donors such as folate [66].

Strengths of this study include a sample size that enabled sex-specific analyses. Several methylation outcomes were sex-specific, highlighting the importance of treating sex as a biological variable. Further, the longitudinal design of the Newborn Epigenetics Study (NEST), where exposure was assessed at enrollment (mean gestation age 12 weeks) and CpG methylation was measured at birth, resulted in minimized recall variability in exposure. The use of pyrosequencing, which is the gold standard in DNA methylation sequencing, also ensured the robustness of methylation measurements. 

We evaluated associations between either the use of clomifene alone to assist in pregnancy or the use of ART (ET/ICSI) to assist in pregnancy, and offspring methylation of imprinted DMRs. These DMRs control genes involved in multiple developmental pathways implicated in a wide range of developmental disorders when dysregulated. As embryo transfer technology is often preceded by the use of super-ovulatory drugs such as clomifene, differences observed between the use of clomifene only and embryo transfer technology (ET/ICSI) would involve the manipulation of embryo and ex-vivo culture conditions of ovum/embryo in the latter.

A limitation of our study and others is the inability to adequately control for unmeasured confounding. For example, maternal education may approximate a wide range of stressors that include, among others, socioeconomic status and access to prenatal care information, which may contribute to variability in offspring methylation status, etc. [67,68]. In our cohort, the use of ART varied widely based on education status, with most mothers who used ART having at least some college education. By removing all mothers who did not have at least some college education, we attempted to isolate ART as an exposure, instead of using education as a proxy for multiple behaviors and exposures.

Another limitation is a lack of data on differences in the media used in in-vitro procedures. The media vary by clinic, even including differences in whether folate (a classical methyl donor) is used [64]. More importantly, in nearly all human studies of ART and offspring outcomes, a recurring interpretation issue arises in disentangling underlying maternal biology-related and ART-related offspring outcomes. That is no different in this study. We were unable to determine if mothers requiring the assistance of super-ovulatory drugs and/or the use of ART have underlying biological differences that are the primary driver of observed methylation differences in offspring cord blood. However, work in mice has suggested that ART does indeed result in changes in offspring epigenetics that are not related to maternal infertility [69]. In further support of maternal biology-independent effects of ART on epigenetic alterations, recent work suggested that in mice, proper imprinting of *Snrpn*, *Kcnq1ot1*, and *H19* is altered by ART but not by age [70]. Further, the underlying biology driving the choice of super-ovulatory drugs vs. more invasive ART procedures may be a contributing factor to the methylation differences seen between the effects of clomifene only vs. ART. These include both maternal and paternal contributions to infertility.

## 4. Materials and Methods

### 4.1. Study Participants

Participants were selected from the prospective Newborn Epigenetics Study (NEST). Enrollment in NEST is described elsewhere [40,71]. Briefly, pregnant women who visited one of six prenatal clinics in Durham, NC, USA and neighboring counties for prenatal care between 2005 and 2011 were targeted for inclusion. Prospective participants who were aged ≥18 years, intended to maintain residence in the area until delivery, and intended to deliver at either Durham Regional Hospital or Duke University Obstetrics Hospital were enrolled. The latter criterion facilitated the collection of parturition data and specimens, including fetal blood from the umbilical cord vein and maternal blood. Of the 3545 women approached, 2546 were enrolled. Participation was not associated with educational level, maternal age, or maternal BMI although Asians were less likely to participate [40,71]. All participants provided informed consent, and the study was approved by Duke University’s institutional review board (Pro00014548; IRB# 7041-07-3R2ER; March 2007). 

### 4.2. Data Collection

#### 4.2.1. Outcome Assessment

For methylation analysis of cord blood, NEST participants were selected based on the availability of recent follow-up data at the time of sample analysis, consent to epigenetic analyses at enrollment, and additional consent to follow-up epigenetic analysis for offspring from ages 3–5 years during follow-up visits. Methylation was assessed in at least one ICR for 1296 participants. Procedures for specimen collection, handling, and methylation are described elsewhere [27,40,71]. Briefly, DNA for pyrosequencing was extracted from offspring cord-blood buffy coat using Puregene (Qiagen, Germantown, MD) reagents. Pyrosequencing was performed on a Pyromark Q96 Pyrosequencer (Qiagen, Germantown, MD). Primers and PCR conditions are described in detail elsewhere [11,72]. The methylation fraction for each CpG dinucleotide was calculated using PyroQ CpG Software (Qiagen, Germantown, MD). The methylation fraction was analyzed at multiple CpGs of DMRs (n) in the ICRs of *H19* (4), *IGF2* (3), *MEG3-IG* (4), *MEG3* (8), *MEST* (4), *PEG3* (10), *PLAGL1* (6), *NNAT* (3), and *SGCE/PEG10* (6). Due to quality control methods, minor differences in sample sizes exist depending on the CpG. Note that these regions are considered established ICRs and are referred to as such in the text. 

#### 4.2.2. Exposure Assessment

We defined exposure groups for ART and clomifene using data from a questionnaire completed by 2209 NEST participants at enrollment. Nearly all participants who used clomifene and/or underwent ART were non-smokers and had at least some college education. Due to known offspring methylation changes from maternal smoking [73,74,75] and social adversity [76,77,78] and because nearly all NEST participants who underwent ART were both college-educated and non-smokers, we excluded from the analysis participants who smoked or had no college education (exposed, *n* = 5 and not exposed, *n* = 717), leaving 574 mother/offspring pairs. NEST participants were asked questions about medications used to assist conception in only one of the two enrollment waves, resulting in different denominators for the ART and clomifene exposure groups (Figure 1).

For ART, we classified participants who answered yes to “Did you use in vitro fertilization or embryo transfer (IVF/ET)? (Yes, No)” or “Did you use intracytoplasmic sperm injection? (Yes, No)” as exposed (*n* = 27). We classified participants who answered no to both questions as non-exposed (*n* = 516). We further excluded from the analysis nine participants who indicated they used ART in the index pregnancy but did not answer questions about whether they underwent ART/ET or ART/ICSI and 22 participants who reported clomifene use without ART exposure. 

For clomifene, we excluded from the analysis 235 of the 574 participants who answered questions about ART during the index pregnancy but did not complete a follow-up questionnaire on medications used to assist conception. Of the 339 remaining participants, we excluded 14 who were in the ART-exposed group. We classified 22 who answered yes to the use of clomifene/Serophene as exposed and the remaining 303 as non-exposed. 

Information on the reasons for ART or clomifene use or how many cycles/attempts were conducted was not available.

### 4.3. Statistical Analyses

We assessed associations of clomifene- and/or ART-conceived offspring with percent methylation of 48 CpGs in nine ICRs using complete-case linear regression stratified by offspring sex. We adjusted models for the year/plate batch of methylation assessment, parity (first child vs. other), maternal BMI at last menstrual period, maternal age, diabetes status (none vs. any of type 1, type 2, or gestational), and race. We sex-stratified the analyses because prior work has demonstrated sex-specific differential methylation patterns of imprinted genes from maternal exposures [79]. Frequencies and percentages of demographics and covariates are reported using non-missing data, which vary by variable. Although extensive quality control has been performed for the pyrosequencing data [11], we conducted an additional analysis by removing outliers based on mean +/− 3 standard deviations. All analyses were conducted in R version 4.0.3 (Vienna, Austria). 

### 4.4. Nomenclature

When discussing ICRs and DMRs, there is often a degree of conflation. An imprint control region (ICR) refers to the CpG sites within a given stretch of DNA sequence that exhibit a uniform, binary pattern of methylation in sperm versus oocyte, and that contributes to the establishment of parent-of-origin-dependent expression in the resulting somatic tissues for the associated gene(s). Throughout the paper, we use the phrase “differentially methylated region” (DMR) to refer to the CpGs, or subsets of CpGs, within ICRs that are known to contribute to regulating the parent-of-origin-dependent monoallelic expression of the imprinted gene(s) controlled by that ICR.

## 5. Conclusions

Despite the limitations detailed in the discussion, with limited candidate epigenetic locus interrogation, our study provides additional evidence to the growing body of literature (reviewed in [80]) demonstrating a relationship between ART and altered offspring methylation in ICRs. When taken together, our data suggest a perturbation of methylation marks at known ICRs. These findings also highlight the importance of treating sex as a biological variable. Further, our study suggests the need for additional research on methods of inducing and collecting ovum as well as the effects of the treatment conditions of ART use on offspring methylation alteration in offspring. Lastly, we highlight the critical need for larger, better-powered studies that can utilize epigenome-wide interrogation in an offspring-sex-dependent manner.

## Figures and Tables

**Figure 1 ijms-23-10450-f001:**
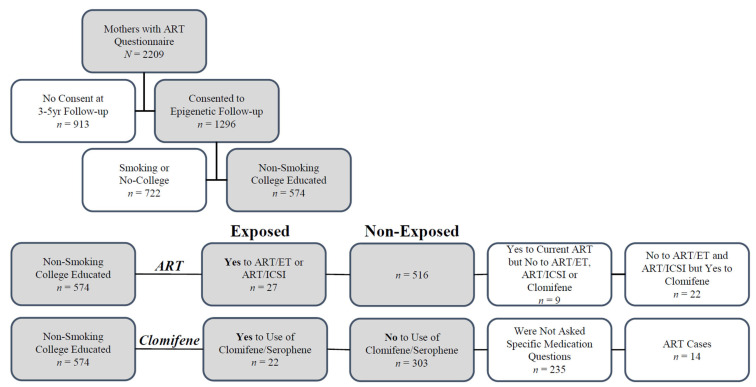
**Cohort Selection and Exposed/Non-exposed Definition:** Mothers from two waves of enrollment were asked questions concerning index pregnancy and use of ART use. Of these participants, 1296 mothers agreed to epigenetic follow-up, including pyrosequencing of offspring cord blood. Cases were overwhelmingly non-smoking and college-educated participants; mothers who either smoked or were not college-educated were excluded (removed 5 cases and 717 controls). For the **ART-exposed** group, 27 cases were defined based on an affirmative answer to either of one of the following questions: “Did you use in vitro fertilization or embryo transfer (IVF/ET)? (Yes, No)” or “Did you use intracytoplasmic sperm injection? (Yes, No)”. There were 9 mothers who answered affirmative answer to “Was current pregnancy conceived with ART? (Yes, No, Don’t Know, Refused)”, but answered negative to the prior two questions and were removed from the analysis. To examine **clomifene-only use** separately, 22 participants who reported use of clomifene/Serophene, but gave negative answers to the questions concerning ART/ET and ART/ICSI were excluded from the ART analysis and included as cases in the clomifene group defined below. For the clomifene group, of the 516 ART controls, 339 mothers answered follow-up questions regarding their use of specific fertility medication. Of these, 22 were classified as cases based on reported use of clomifene/Serophene, 14 were excluded from the analysis because they were included as ART cases, and the remainder were classified as controls (*n* = 303).

**Figure 2 ijms-23-10450-f002:**
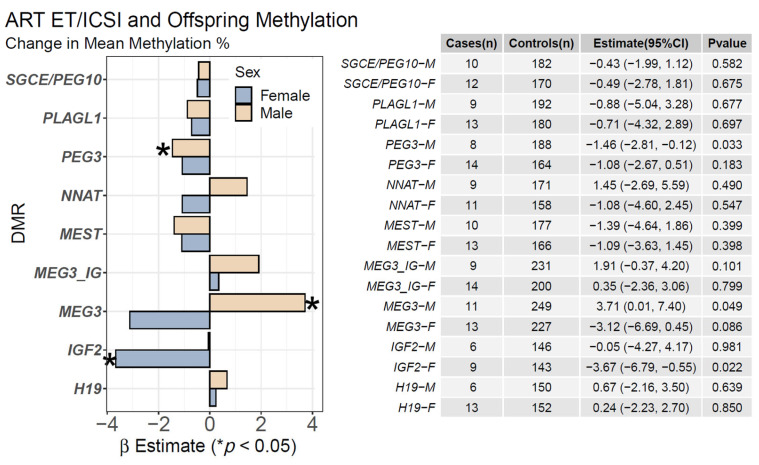
**Associations of ART ET/ICSI and Mean CpG Methylation:** We assessed the relationship between ART use for the index pregnancy and mean CpG methylation at nine DMRs using linear regression. We stratified models by sex and adjusted for pyrosequencing batch, pre-pregnancy maternal BMI, race, maternal age, diabetes, and parity. Estimates (β) are shown for males (tan) and females (blue) and represent the mean percentage of methylation change associated with maternal use of ART (* *p* < 0.05).

**Figure 3 ijms-23-10450-f003:**
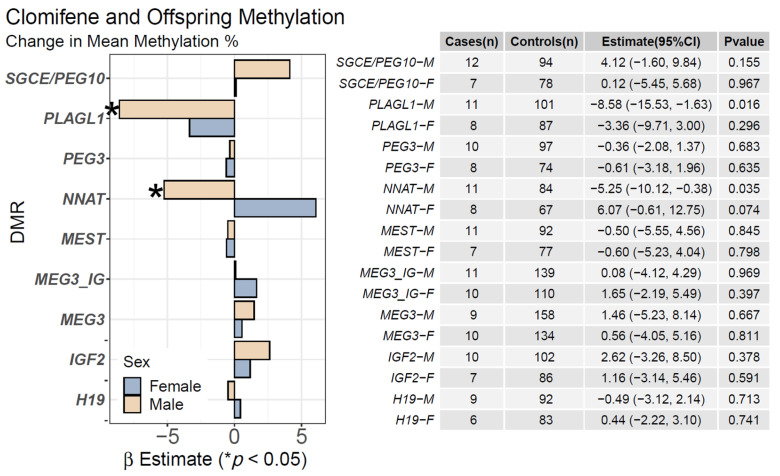
**Associations of Clomifene and Mean CpG Methylation:** We assessed the relationship between the use of clomifene only and mean CpG methylation at nine DMRs using linear regression. We stratified models by sex and adjusted for pyrosequencing batch, maternal pre-pregnancy BMI, race, maternal age, diabetes, and parity. Estimates (β) are shown for males (tan) and females (blue) and represent the mean percentage of methylation change associated with maternal use of clomifene without ART (* *p* < 0.05).

**Table 1 ijms-23-10450-t001:** Participant demographics.

Demographic		ART ET/ICSI		
		Mean (SD) or Frequency (% of Total)		
		Exposed	Non-Exposed	All
		*n* = 27 (4.7%)	*n* = 516 (89.9%)	*n* = 543 (100%)
Maternal age ^#^		36.3 (3.6)	30.1 (5.5)	30.4 (5.5)
Maternal pre-pregnancy BMI ^#^		24.5 (6.1)	27.7 (8.8)	27.5 (8.7)
Race / Ethnicity ^#^	White	21 (77.8%)	242 (46.9%)	263 (48.4%)
	Non-white	6 (22.2%)	274 (53.1%)	280 (51.6%)
Parity *	Nulliparous	17 (63%)	213 (41.4%)	230 (42.5%)
	One or more	10 (37%)	301 (58.6%) *	311 (57.5%) *
Diabetes ^†^	None	25 (96.2%)	444 (88.1%)	469 (88.5%)
	Any type	1 (3.8%)	60 (11.9%)	61 (11.5%)
Sex	Male	11 (40.7%)	272 (52.7%)	283 (52.1%)
	Female	16 (59.3%)	244 (47.3%)	260 (47.9%)
**Demographic**		**Clomifene Only**		
		***n* = 22 (6.8%)**	***n* = 303 (93.2%)**	***n* = 325 (100%)**
Maternal age ^#^		33.6 (4.9)	30.0 (5.1)	30.2 (5.2)
Maternal pre-pregnancy BMI		25.6 (5.8)	27.4 (8.1)	27.2 (7.9)
Race/Ethnicity ^#^	White	18 (81.8%)	143 (47.2%)	161 (49.5%)
	Non-white	4 (18.2%)	160 (52.8%)	164 (50.5%)
Parity	Nulliparous	7 (31.8%)	136 (45.0%)	143 (44.0%)
	One or more	15 (68.2%)	167 (55.0%)	182 (55.0%)
Diabetes ^‡^	No	20 (90.9%)	269 (90.0%)	289 (90.0%)
	Any type	2 (9.1%)	30 (10.0%)	32 (10.0%)
Sex	Male	12 (54.5%)	165 (54.5%)	177 (54.5%)
	Female	10 (45.5%)	138 (45.5%)	148 (45.5%)

Missing covariates: * Two non-exposed; ^†^ One exposed and 12 non-exposed; ^‡^ Four non-exposed. ^#^ Exposed and non-exposed means (continuous variables—Student *t*-test) or expected frequencies (categorical variables − chi-squared test) differ (*p* < 0.05).

## Data Availability

The data underlying the analysis outlined in the manuscript may be available on request from the corresponding author. The data are not publicly available to protect the privacy of study participants.

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
