# Peer review of "Clomifene and Assisted Reproductive Technology in Humans Are Associated with Sex-Specific Offspring Epigenetic Alterations in Imprinted Control Regions"

_ijms, 2022, doi:10.3390/ijms231810450_

Round 1

Reviewer 1 Report

Minor comments, see pdf attached.

Author Response

Dear Reviewer,

Thank you for your extensive feedback and suggestions. We have responded in kind to your comments in the .PDF you provided. Please find it attached. Because of your review, we think the revised manuscript is a better, clearer manuscript.

Kind regards,

John House

Reviewer 2 Report

The authors investigate the offspring epigenetic changes after the mother take clomifene or Assisted Reproductive Technology (ART). The aim is interesting but some questions need to answer before publication.

1. Figure 1 is not clear. Please provide a high-resolution figure.

2. Table 1 should add statistics on the significance if it is possible.

3. The importance of those genes, MEST and NNAT, should be descript in the manuscript.

4. The core blood is fresh or stored in the refrigerator when methylation assay? If the core blood storage for a long time can affect the methylation result?

5. The authors thought that epigenetic changes related to ART procedures or medicine taking. But mothers can inherit their children including epigenetic changes. Is it possible that the epigenetic change of offspring is from the mother's?

6. If the authors can provide more data to support the results, the results will be more solid.

Author Response

Dear Reviewer,

Thank you for you feedback. We have modified the manuscript where indicated and have responded to each of your points below. The manuscript is better and clearer because of your input. 

Kind regards,

John House

1. Figure 1 is not clear. Please provide a high-resolution figure.

Thank you for the suggestion. We have revised Figure 1 based on this feedback and the feedback from Reviewer 1. Although we have pasted a new figure into the Word Document, Word has still made it a little blurry.  A complete high-rez figure in true-type font will be provided to IJMS. 

2. Table 1 should add statistics on the significance if it is possible.

We thank you for the suggestion. The table was too full to add another column with the manuscript guidelines on width, but we did add the following to the table to demonstrate p < 0.05 between exposed and non-exposed covariates.  "#Exposed and non-exposed means (continuous variables – Student t test) or expected frequencies (categorical variables – chi-squared test) are different (p < 0.05)"

3. The importance of those genes, MEST and NNAT, should be descript in the manuscript.

Thank you for pointing out this omission. We have added to the discussion relevant work showing the importance of these genes in previous research concerning ART and pointed out how are results are either similar or different.

4. The core blood is fresh or stored in the refrigerator when methylation assay? If the core blood storage for a long time can affect the methylation result?

These data were generated from umbilical cordblood collected at birth via umbilical vein puncture, centrifuged and immediately stored at -80C until thawed, bisulfite converted, and pyrosequenced. Storage at -80 is minimal, and any degradation should occur similarly between controls and cases.  

5. The authors thought that epigenetic changes related to ART procedures or medicine taking. But mothers can inherit their children including epigenetic changes. Is it possible that the epigenetic change of offspring is from the mother's?

Thank you for the question. It is true that some epigenetic marks are heritable. However, imprinted marks, where one allele is parentally silenced tend to be more stable. In addition, we actually are assessing changes in heritability if you will by assessing methylation changes between mothers who did and did not use ART/Clomifene. One shortfall in all studies of ART is the difficulty in ascertaining if offspring outcomes were because of ART or because of underlying infertility of the parents. This is discussed thoroughly in the discussion. 

6. If the authors can provide more data to support the results, the results will be more solid.

Thank you for the suggestion. We are limited in more analyses due to the special issue time requirements. We have redone and further clarified the results and supplemental figures per reviewer 1's suggestion.

Reviewer 3 Report

In this manuscript, Lloyd et al. examined the effect of ovulation drugs and assisted reproductive technology (ART) on CpG methylation in differentially methylated CpGs in known imprint control regions (ICRs) in a retrospective North Carolina-based birth cohort. They found associations between ART and hypomethylation of the PEG3 DMR. They also found that a male-specific association of ART with hypermethylation of the MEG3 DMR in males and a female-specific association of ART with hypomethylation of the IFG2 DMR.

The authors performed pyrosequencing of 48 CpG sites in nine ICRs. This reviewer has the following concerns.

Major comments:

1. Adverse maternal and offspring outcomes are missing. Do the offsprings having aberrant methylations show cardiovascular disease and metabolic and behavioral outcomes? It is not able to be analyze the association between aberrant methylation and adverse maternal and offspring outcomes.

2. Aberrant methylation identified here is consistent with a previous report. However, this reviewer wonder whether these aberrant methylation is reproductively validated in other independent samples.

3. In Figure 1, number of exclusions should be described. For example, for ART, among 574 participants, 27 is exposed, 516 is non-exposed, 9 is excluded, and 22 is reported to be clomifene use without ART exposure. For clomifene, exclusion of 235 participants and exclusion of 14 should be described. Although it is described in 2.2.2.Exposure assessment.

Author Response

Dear Reviewer 3,

Thank you for your comments and feedback. We have addressed them below and in the manuscript where indicated. Thank you for your time.  We also made extensive edits to the manuscript and analyses requested by the other reviewers. Because of your reviews, the manuscript is much improved.

Kind regards,

John House

1. Adverse maternal and offspring outcomes are missing. Do the offsprings having aberrant methylations show cardiovascular disease and metabolic and behavioral outcomes? It is not able to be analyze the association between aberrant methylation and adverse maternal and offspring outcomes.

This is a great question, but not one that we have data to answer. This cohort was not followed long enough to gather CVD information. While we do have some behavioral assessment, it is on a much smaller portion of the respondents. The frequency of ART in those individuals prevent meaningful analysis.

2. Aberrant methylation identified here is consistent with a previous report. However, this reviewer wonder whether these aberrant methylation is reproductively validated in other independent samples.

We tried to find a replication cohort without success prior to this invited submission to the special edition. Most other epigenetic work has been chip based (i.e. Illumina 450k/850k) and the overlap of the 48 CpGs from these 9 ICRs is only 3, and those 3 were not in our significant findings. What we have done, is to see how our data overlap with other studies examining ART and imprinted genes, and as you mentioned, they are largely consistent and build on the body of evidence suggesting a possible mechanism by which children concieved with ART have increased adverse health outcomes. Our work also highlights the need for more work examining differences between ART methods as well as media constituents.

3. In Figure 1, number of exclusions should be described. For example, for ART, among 574 participants, 27 is exposed, 516 is non-exposed, 9 is excluded, and 22 is reported to be clomifene use without ART exposure. For clomifene, exclusion of 235 participants and exclusion of 14 should be described. Although it is described in 2.2.2.Exposure assessment.

Thank you for this suggestion. We have modified the intext legend as suggested.

Round 2

Reviewer 2 Report

It's OK. Thank you.

Reviewer 3 Report

All the comments have been addressed.